# A Diffusion Weighted Graph Framework for New Intent Discovery

**Wenkai Shi**[1], **Wenbin An**[1], **Feng Tian**[2*], **Yan Chen**[2], **Qinghua Zheng**[2]
**QianYing Wang**[3], **Ping Chen**[4]

[1] School of Automation Science and Engineering, Xi'an Jiaotong University
[2] School of Computer Science and Technology, MOEKLNNS Lab, Xi'an Jiaotong University
[3] Lenovo Research [4] Department of Engineering, University of Massachusetts Boston
shiyibai778@gmail.com, {fengtian,chenyan}@mail.xjtu.edu.cn
wenbinan@stu.xjtu.edu.cn, wangqya@lenovo.com, ping.chen@umb.edu

## Abstract

New Intent Discovery (NID) aims to recognize both new and known intents from unlabeled data with the aid of limited labeled data containing only known intents. Without considering structure relationships between samples, previous methods generate noisy supervisory signals which cannot strike a balance between quantity and quality, hindering the formation of new intent clusters and effective transfer of the pre-training knowledge. To mitigate this limitation, we propose a novel *Diffusion Weighted Graph Framework* (DWGF) to capture both semantic similarities and structure relationships inherent in data, enabling more sufficient and reliable supervisory signals. Specifically, for each sample, we diffuse neighborhood relationships along semantic paths guided by the nearest neighbors for multiple hops to characterize its local structure discriminately. Then, we sample its positive keys and weigh them based on semantic similarities and local structures for contrastive learning. During inference, we further propose *Graph Smoothing Filter* (GSF) to explicitly utilize the structure relationships to filter high-frequency noise embodied in semantically ambiguous samples on the cluster boundary. Extensive experiments show that our method outperforms state-of-the-art models on all evaluation metrics across multiple benchmark datasets. Code and data are available at https://github.com/yibai-shi/DWGF.

## 1 Introduction

Even though current machine learning methods have achieved superior performance on many NLP tasks, they often fail to meet application requirements in an open-world environment. For instance, general intent classification models trained on predefined intents cannot recognize new intents from unlabeled dialogues, which is a clear obstacle for real-world applications. Therefore, research on

---

*Corresponding Authors.

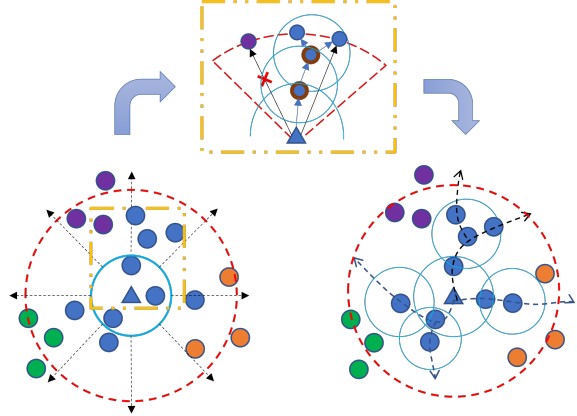

Figure 1: Illustration of the transformation of supervisory signal generation method. **Bottom Left**: generating supervisory signals indiscriminately along all directions of the hypersphere, which is sensitive to threshold changing. **Top**: an example of selecting samples with semantic paths. **Bottom Right**: generating supervisory signals directionally with structure relationships composed of multiple semantic paths in a relaxed feature hypersphere.

New Intent Discovery (NID), which aims to discover new intents from unlabeled data automatically, has attracted much attention recently.

Most existing NID methods (Lin et al., 2020; Zhang et al., 2021; Wei et al., 2022; Zhang et al., 2022; An et al., 2023) adopt a two-stage training strategy: pre-training on labeled data, then learning clustering-friendly representation with pseudo supervisory signals. However, previous methods only rely on semantic similarities to generate supervisory signals based on the assumption that samples within the feature hypersphere belong to the same category as the hypersphere anchor, e.g. cluster centroids (Zhang et al., 2021), class prototypes (An et al., 2022b), or query samples (Zhang et al., 2022).

Even though these methods can learn some discriminative features, they still face limitations in generating both adequate and reliable supervisory

signals, which we call the **Quantity and Quality Dilemma**. Specifically, as shown in Fig.1 Bottom Left, these methods rely on a fixed threshold to determine the search radius of the hypersphere. Shrinking the threshold (blue solid line) helps retrieve more accurate positive keys, but it loses information from positive keys out of the hypersphere, resulting in a low recall. However, simply relaxing the threshold (red dashed line) will introduce much noise and lead to low accuracy.

Quantity and Quality Dilemma is caused by the fact that the previous methods searched positive keys indiscriminately along all directions of the hypersphere with a fixed search radius. In order to selectively sample both adequate and reliable positive keys to ensure the formation of new intent clusters, we propose to model and utilize **structure relationships** inherent in data, which reflect the semantic correlations between samples from the perspective of connectivity. As shown in Fig.1 Top, for each sample, we first initialize its $k$-nearest neighbors with a tightened threshold. Then we connect any two samples if they have at least one shared neighbor since the semantics of the shared neighbor are highly correlated with the samples on both sides. According to this rule, we identify two samples (with brown borders in Fig.1 Top) that can be used as bridges and diffuse the anchor along them to search positive keys near the boundary of the hypersphere, forming the final semantic path. In the case of the same semantic similarity, we additionally require the positive keys to appear on the semantic paths diffused from the anchor.

In this paper, we propose a novel Diffusion Weighted Graph Framework to model and utilize structure relationships. Specifically, from any anchor, we diffuse neighborhood relationships along the nearest neighbor-guided semantic paths for multiple hops to construct the final DWG. As shown in Fig.1 Bottom Right, then we sample positive keys along the semantic paths (arrow lines) in DWG within the relaxed feature hypersphere. Moreover, sampled keys are assigned to different contrastive weights according to their frequency of being sampled on different semantic paths, where keys that are diffused repeatedly from different outsets will accumulate larger values and vice versa. We conduct contrastive learning with sampled positive keys and corresponding weights in the embedding space. Apart from considering the sample-sample structure relationships from the local view, we

adopt the idea of Xie et al. (2016) to help learn clustering-friendly representations from the global view through self-training.

During the inference stage, in order to filter high-frequency noise embodied in the semantically ambiguous samples on the cluster boundary, we propose a novel inference improvement *Graph Smoothing Filter* (GSF), which utilizes normalized graph Laplacian to aggregate neighborhood information revealed by structure relationships of testing samples. Smoothed testing features help to obtain better clustering results.

Our main contributions can be summarized as follows:

- We propose a Diffusion Weighted Graph Framework (DWGF) for NID, which can capture both semantic similarities and structure relationships inherent in data to generate adequate and reliable supervisory signals.

- We improve inference through Graph Smoothing Filter (GSF), which exploits structure relationships to correct semantically ambiguous samples explicitly.

- We conduct extensive experiments on multiple benchmark datasets to verify the effectiveness.

## 2 Related Work

### 2.1 New Intent Discovery

Semi-supervised NID aims to discover novel intents by utilizing the prior knowledge of known intents. First, it is assumed that the labeled data and the unlabeled data are disjoint in terms of categories. To tackle the NID challenge under this setting, Mou et al. (2022a) proposed a unified neighbor contrastive learning framework to bridge the transfer gap, while Mou et al. (2022b) suggested a one-stage framework to simultaneously classify novel and known intent classes. However, a more common setting in practice is that the unlabeled data are mixed with both known and new intents. Compared to the previous setting, the latter is more challenging because the above methods have difficulty distinguishing a mixture of two kinds of intents and are prone to overfit the known intent classes. To this end, Lin et al. (2020) conducted pair-wise similarity prediction to discover novel intents, and Zhang et al. (2021) used aligned pseudo-labels to help the model learn clustering-friendly representations. Recently, contrastive learning has

become an important part of NID research. For example, An et al. (2022a) proposed hierarchical weighted self-contrasting to better control intra-class and inter-class distance. Wei et al. (2022) exploited supervised contrastive learning (Khosla et al., 2020) to pull samples with the same pseudo-label closer. An et al. (2022b) achieved a trade-off between generality and discriminability in NID by contrasting samples and corresponding class prototypes. Zhang et al. (2022) acquired compact clusters with the method of neighbor contrastive learning. However, these methods don't fully explore the structure relationships inherent in data, causing the generated supervisory signals to fall into a Quantity and Quality Dilemma.

## 2.2 Contrastive Learning

Contrastive learning pulls similar samples closer, pushes dissimilar samples far away, and has gained promising results in computer vision (Chen et al., 2020; He et al., 2020; Khosla et al., 2020) and natural language processing (Gao et al., 2021; Kim et al., 2021). Inspired by the success of contrastive learning, a large number of works extend the definition of positive and negative keys in it to adapt to more research fields. For example, Li et al. (2021) conducted cluster-level contrastive learning in the column space of logits, Li et al. (2020) proposed to use cluster centroids as positive keys in contrastive learning, and Dwibedi et al. (2021) treated nearest neighbors in feature space as positive keys. These works all help model to learn cluster-friendly representations that benefit NID. However, they solely rely on semantic similarities to search positive keys, which inevitably generate noisy pseudo supervisory signals.

## 3 Methods

### 3.1 Problem Formulation

Traditional intent classification task follows a closed-world setting, i.e., the model is only developed based on labeled dataset $\mathcal{D}^l = \{(x_i, y_i)|y_i \in \mathcal{Y}^k\}$, where $\mathcal{Y}^k$ refers to the set of known intent classes. New Intent Discovery follows an open-world setting, which aims to recognize all intents with the aid of limited labeled known intent data and unlabeled data containing all classes. Therefore, in addition to the above $\mathcal{D}^l$, $\mathcal{D}^u = \{x_i|y_i \in \mathcal{Y}^k \cup \mathcal{Y}^n\}$ from both known intents $\mathcal{Y}^k$ and new intents $\mathcal{Y}^n$ will be utilized to train the model together. Finally, the model performance will be evaluated on the testing set $\mathcal{D}^t = \{x_i|y_i \in \mathcal{Y}^k \cup \mathcal{Y}^n\}$.

### 3.2 Approach Overview

Fig.2 illustrates the overall architecture of our proposed Diffusion Weighted Graph Framework. The framework includes two parts: training with Diffusion Weighted Graph (DWG) and inference with Graph Smoothing Filter (GSF). Firstly, we conduct pre-training detailed in Sec.3.3. Secondly, as shown in Fig.2's **I**, we extract intent representations to simultaneously conduct self-training from the global view and contrastive learning with DWG from the local view. More training details are provided in Sec.3.4. Finally, as shown in Fig.2's **II**, we construct GSF to smooth testing features and adopt KMeans clustering to complete the inference. More inference details are provided in Sec.3.5.

In summary, combined with structure relationships, our proposed DWGF can 1) break through the limitation of tightened threshold and achieve higher sampling accuracy and recall simultaneously; 2) suppress sampling noise while retaining rich semantics through soft weighting; 3) consider the local sample-sample supervision and the global sample-cluster supervision simultaneously; 4) filter high-frequency noise embodied in semantically ambiguous samples on the cluster boundary during inference.

### 3.3 Model Pre-training

We use BERT (Devlin et al., 2019) to encode input sentences and take all token embeddings from the last hidden layer. Then we apply average pooling to acquire the final intent representations.

$$z_i = mean\text{-}pooling(BERT(x_i)) \qquad (1)$$

where $x_i$ and $z_i$ refer to $i\text{-}th$ input sentence and corresponding representation. Motivated by (Zhang et al., 2021), we use Cross-Entropy loss on labeled data to acquire prior knowledge from known intents. Furthermore, we follow (Zhang et al., 2022) to use Masked Language Modeling (MLM) loss on all training data to learn domain-specific semantics. We pre-train the model with the above two kinds of loss simultaneously:

$$\mathcal{L}_{pre} = \mathcal{L}_{ce}(\mathcal{D}^l) + \mathcal{L}_{mlm}(\mathcal{D}^u) \qquad (2)$$

where $\mathcal{D}^l$ and $\mathcal{D}^u$ are labeled and unlabeled dataset, respectively.

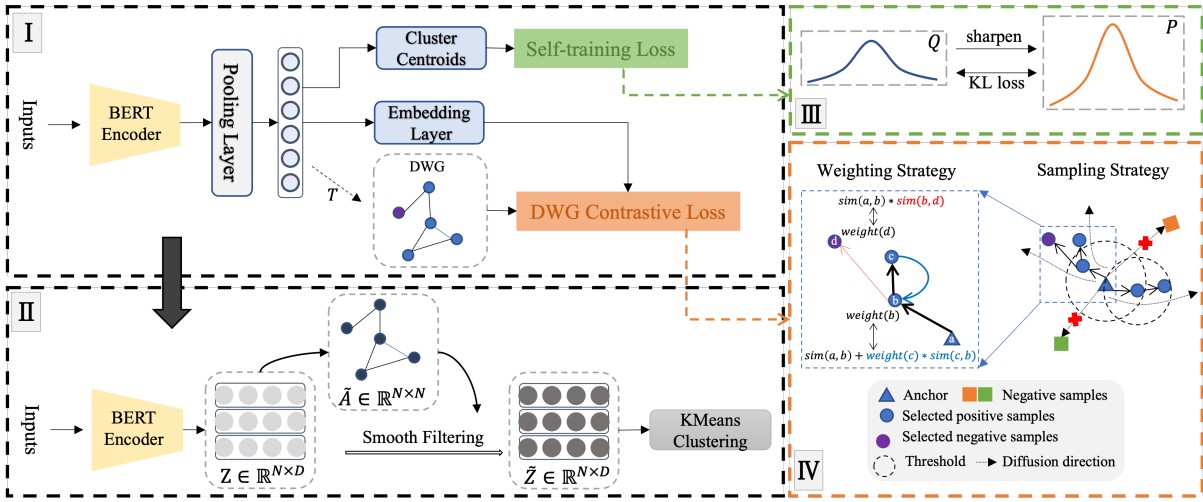

Figure 2: Overall architecture of our proposed Diffusion Weighted Graph Framework (DWGF). **I**: illustration of the model training. **II**: illustration of the inference. **III**: illustration of self-training. **IV**: illustration of contrastive learning based on DWG.

### 3.4 Representation Learning with DWG

After pre-training, we extract all training samples' $l2$-normalized intent representations and initialize the instance graph $A_0$ with a monomial kernel (Iscen et al., 2017) as the similarity metric.

$$A_{ij}^0 := \begin{cases} max(z_i^T z_j, 0)^\rho, & i \neq j \wedge j \in \mathcal{N}_k(z_i) \\ 0, & otherwise \end{cases}$$ (3)

Here $\mathcal{N}_k(z_i)$ saves the indices of $k$-nearest neighbors of $z_i$, and $\rho$ controls the weights of similarity. We set $\rho$ to 1 for simplicity and generality.

Different from Zhang et al. (2022), we reduce the neighborhood size and retain the similarities with anchor instead of 0-1 assignment. We aim to model the structure relationships through KNN rather than directly sample positive keys. With the initial high-confidence neighbors, we perform subsequent diffusion to complete the DWG and implement sampling and weighting.

**Sampling Strategy**. As shown in Fig.2's **IV**, the smaller neighborhood size first ensures that semantically unrelated heterogeneous samples are not used as outsets for diffusion. Then we start with the anchor and diffuse its neighborhood relationships along semantic paths guided by high-confidence neighbors, which would be included as new anchors for the next diffusion. We define DWG as the accumulation of multiple self-multiplications of $A_0$.

$$\hat{A} = \sum_{i=1}^{r} \theta^{i-1} \cdot A_0^i$$ (4)

where $r$ refers to the diffusion rounds, $\theta$ is the magnitude of diffusion, which is set it to 1 for simplicity. Combined with a relaxed semantic similarity threshold $\gamma$, we further filter keys in DWG with similarity below the threshold, i.e. $\hat{A}_{ij} = 0$ if $z_i^T z_j < \gamma$.

**Weighting Strategy**. Apart from semantic similarity, DWG $\hat{A}$ also reflects the confidence of sampled keys to the anchor from the perspective of the frequency that the key is repeatedly diffused. However, the numerical scale of each row in $\hat{A}$ varies significantly due to the different diffusion process. To ensure consistency in subsequent contrastive learning, we normalize them to $[0, 1]$ interval with the degree $D_i = \sum_j \hat{A}_{ij}^r$ of $x_i$ and modulation factor $\lambda$.

$$w_{ij} = \begin{cases} min(1, \lambda \cdot \frac{\hat{A}_{ij}}{D_i}), & \hat{A}_{ij} > 0 \\ 0, & \hat{A}_{ij} = 0 \end{cases}$$ (5)

As shown in Fig.2's **IV**, such a soft weighting strategy has two advantages. Firstly, the blue line shows that homogeneous samples will prevail in contrastive learning because of the cumulative influence from the multiple diffusion of different keys. Secondly, the red line indicates that even if heterogeneous samples are selected, they will be assigned smaller weights because of infrequent sampling and lower similarity to the diffusion outset.

To fully utilize sampled positive keys, we maintain a momentum encoder and a dynamic queue following (He et al., 2020), which can help the model benefit from contrasting large amounts of

consistent keys at once. At the end of each iteration, the dynamic queue will be updated by adding current samples and removing the oldest samples. We denote the final DWG contrastive learning loss as:

$$\mathcal{L}_{local} = -\frac{1}{BN} \sum_{i=1}^{B} \sum_{j=1}^{N} \log \frac{w_{ij} \cdot e^{h_i^T \cdot \tilde{h}_j / \tau}}{\sum_{j'=1}^{N} e^{h_i^T \cdot \tilde{h}_{j'} / \tau}} \tag{6}$$

where $B$ and $N$ refer to the size of the batch and dynamic queue, respectively. $h_i$ is the embedding of $x_i$. $\tilde{h}_j$ is the embedding of $x_j$ stored in the dynamic queue. $\tau$ is the temperature hyperparameter.

Apart from considering the sample-sample neighborhood structure from the local view, we adopt the idea of Xie et al. (2016) to add the sample-cluster supervision from the global view. Firstly, we initialize the cluster centroids with the KMeans result on pre-training features and use $t$-distribution to estimate the distance between intent representations $z_i$ and cluster centroid $\mu_k$:

$$Q_{ik} = \frac{(1 + \|z_i - \mu_k\|^2 / \eta)^{-\frac{\eta+1}{2}}}{\sum_{k'} (1 + \|z_i - \mu_{k'}\|^2 / \eta)^{-\frac{\eta+1}{2}}} \tag{7}$$

Here we set $\eta = 1$ for simplicity. Then we generate the auxiliary distribution with both instance-wise and cluster-wise normalization:

$$P_{ik} = \frac{Q_{ik}^2 / f_k}{\sum_{k'} Q_{ik'}^2 / f_k'} \tag{8}$$

where $f_k = \sum_i Q_{ik}$ are soft cluster frequencies. Finally, the cluster assignment distribution $Q$ is optimized by minimizing KL-divergence with the corresponding auxiliary distribution $P$:

$$\mathcal{L}_{global} = \frac{1}{B} \sum_{i=1}^{B} \sum_{k=1}^{|\mathcal{Y}|} P_{ik} log \frac{P_{ik}}{Q_{ik}} \tag{9}$$

Overall, the training objective of our model can be formulated as follows:

$$\mathcal{L} = \mathcal{L}_{local} + \alpha * \mathcal{L}_{global} \tag{10}$$

where $\alpha$ is the relative weight of self-training loss.

### 3.5 Inference with GSF

During the training phase, we model and utilize the structure relationships to help the encoder learn representations that are aware of the local structures. Therefore, the structure relationships inherent in the testing set can be captured by the trained

| Dataset | $|\mathcal{Y}^k|$ | $|\mathcal{Y}^n|$ | $|\mathcal{D}^l|$ | $|\mathcal{D}^u|$ | $|\mathcal{D}^t|$ |
|---|---|---|---|---|---|
| BANKING | 58 | 19 | 673 | 8330 | 3080 |
| StackOverflow | 15 | 5 | 1350 | 16650 | 1000 |
| CLINC | 113 | 37 | 1344 | 16656 | 2250 |

Table 1: Statistics of datasets. $|\mathcal{Y}^k|$, $|\mathcal{Y}^n|$, $|\mathcal{D}^l|$, $|\mathcal{D}^u|$ and $|\mathcal{D}^t|$ represent the number of known categories, new categories, labeled data, unlabeled data and testing data.

encoder and utilized to improve inference in an explicit way.

Specifically, we extract features of the testing set and construct corresponding instance graph $A_t$ as in Sec.3.4. Then, with the renormalization trick $\tilde{A} = I + A_t$ (Kipf and Welling, 2016), we compute the symmetric normalized graph Laplacian:

$$\tilde{L}_{sym} = \tilde{D}^{-1} \tilde{L} \tilde{D}^{-1} \tag{11}$$

where $\tilde{D}_{ii} = \sum_j \tilde{A}_{ij}$ and $\tilde{L} = \tilde{D} - \tilde{A}$ are degree matrix and Laplacian matrix corresponding to $\tilde{A}$. According to (Wang et al., 2019), we denote the Graph Smoothing Filter (GSF) as:

$$H = (I - 0.5 * \tilde{L}_{sym})^t \tag{12}$$

where $t$ refers to the number of stacking layers. We apply the filter to the extracted features and acquire the smoothed feature matrix $\tilde{Z} = HZ$ for KMeans clustering. To the best of our knowledge, this is the first attempt to apply the structure-based filter to inference in NID.

## 4 Experiments

### 4.1 Datasets

We evaluate our method on three benchmark datasets. **BANKING** (Casanueva et al., 2020) is a fine-grained intent classification dataset. **Stack-Overflow** (Xu et al., 2015) is a question classification dataset collected from technical queries online. **CLINC** released by (Larson et al., 2019) is a multi-domain intent classification dataset. More details of these datasets are summarized in Table 1.

### 4.2 Comparison Methods

We compare our method with various baselines and state-of-the-art methods.

**Unsupervised Methods**. GloVe-KM: KMeans with GloVe embeddings (Pennington et al., 2014); SAE-KM: KMeans with embeddings learned by stacked auto-encoder; DEC: Deep Embedded Clustering (Xie et al., 2016); DCN: Deep Clustering

| Method | BANKING | | | StackOverflow | | | CLINC | | |
|---|---|---|---|---|---|---|---|---|---|
| | NMI | ARI | ACC | NMI | ARI | ACC | NMI | ARI | ACC |
| DeepCluster | 39.72 | 7.78 | 18.93 | 17.52 | 3.09 | 18.64 | 53.82 | 12.27 | 28.46 |
| GloVe-KM | 48.75 | 12.74 | 27.92 | 21.79 | 4.54 | 24.26 | 54.57 | 12.18 | 29.55 |
| SAE-KM | 60.12 | 24.00 | 37.38 | 48.72 | 23.36 | 37.16 | 73.13 | 29.95 | 46.75 |
| DEC | 62.92 | 25.68 | 39.35 | 61.32 | 21.17 | 57.09 | 74.83 | 27.46 | 46.89 |
| DCN | 62.94 | 25.69 | 39.36 | 61.34 | 24.98 | 57.09 | 75.66 | 31.15 | 49.29 |
| DTC | 74.51 | 44.57 | 57.34 | 67.02 | 55.14 | 71.14 | 90.54 | 65.02 | 74.15 |
| CDAC+ | 71.76 | 40.68 | 53.36 | 76.68 | 43.97 | 75.34 | 86.65 | 54.33 | 69.89 |
| DAC | 79.56 | 53.64 | 64.90 | 75.24 | 60.09 | 78.74 | 93.89 | 79.75 | 86.49 |
| DSSCC | 81.24 | 58.09 | 69.82 | 77.08 | 68.67 | 82.65 | 93.87 | 81.09 | 87.91 |
| PTJN | 81.69 | 59.20 | 71.77 | 75.43 | 61.90 | 74.18 | 94.41 | 81.07 | 87.35 |
| DPN | 82.58 | 61.21 | 72.96 | 78.39 | 68.59 | 84.23 | 95.11 | 86.72 | 89.06 |
| DCSC | 84.65 | 64.55 | 75.18 | - | - | - | 95.28 | 84.41 | 89.70 |
| CLNN | 85.77 | 67.6 | 76.82 | 81.62 | 74.74 | 86.6 | 96.08 | 86.97 | 91.24 |
| Ours | 86.41 | 68.16 | 79.38 | 81.73 | 75.30 | 87.6 | 96.89 | 90.05 | 94.49 |

Table 2: Evaluation (%) on testing sets. Average results over 3 runs are reported. We set the known class ratio $|\mathcal{Y}_k|/|\mathcal{Y}_k \cap \mathcal{Y}_n|$ to 0.75, and the labeled ratio of known intent classes to 0.1 to conduct experiments.

| Methods | NMI | ARI | ACC |
|---|---|---|---|
| Ours | 86.41 | 68.16 | 79.38 |
| - GSF | 85.82 | 66.96 | 78.21 |
| - Self-training | 85.78 | 66.77 | 77.73 |
| - DWG | 53.89 | 19.30 | 33.05 |

Table 3: Ablation study on the effectiveness of different components. '-' means that we remove the corresponding component.

Network (Yang et al., 2017); DeepCluster: Deep Clustering (Caron et al., 2018).
**Semi-supervised Methods**. DTC: Deep Transfer Clustering (Han et al., 2019); CDAC+: Constrained Adaptive Clustering (Lin et al., 2020); DAC: Deep Aligned Clustering (Zhang et al., 2021); DSSCC: Deep Semi-Supervised Contrastive Clustering (Kumar et al., 2022); DCSC: Deep Contrastive Semi-supervised Clustering (Wei et al., 2022); DPN: Decoupled Prototypical Network (An et al., 2022b); CLNN: Contrastive Learning with Nearest Neighbors (Zhang et al., 2022); PTJN: Robust Pseudo Label Training and Source Domain Joint-training Network (An et al., 2023). Notably, for a fair comparison, the external dataset is not used in CLNN as other methods.

### 4.3 Evaluation Metrics

We adopt three metrics for evaluating clustering results: Normalized Mutual Information (NMI), Adjusted Rand Index (ARI), and clustering Accu-

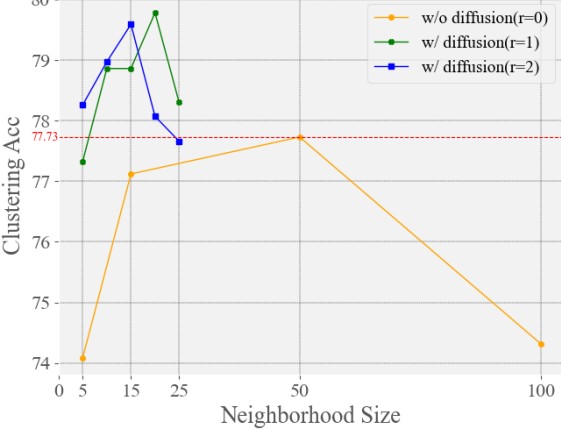

Figure 3: Evaluation (%) under non-diffusion and diffusion-based conditions.

racy (ACC) based on the Hungarian algorithm.

### 4.4 Implementation Details

We use the pre-trained BERT model (bert-bsae-uncased) as our backbone and AdamW optimizer with 0.01 weight decay and 1.0 gradient clipping for parameter update. During pre-training, we set the learning rate to $5e^{-5}$ and adopt the early-stopping strategy with a patience of 20 epochs. During representation learning with DWG, we set the first-order neighborhood size/number of diffusion rounds $k = 15/r = 2$ for BANKING and CLINC, and $k = 50/r = 2$ for StackOverflow to construct DWG, which is updated per 50 epochs. Relaxed threshold $\gamma$, modulation factor $\lambda$, loss weight $\alpha$

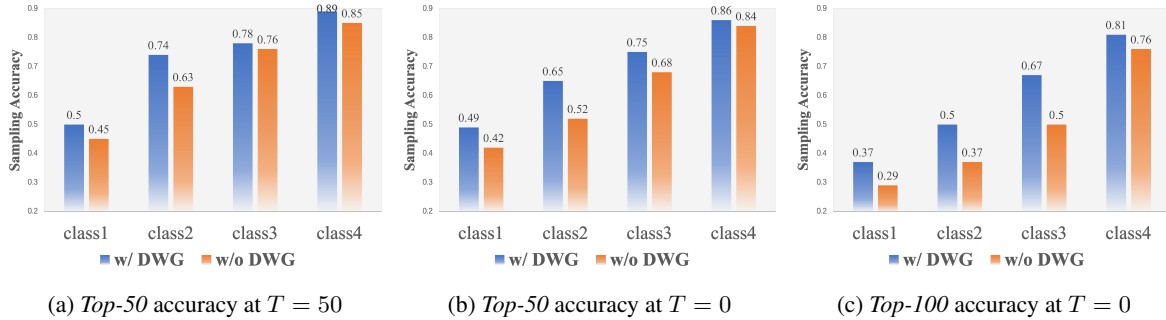

(a) *Top-50* accuracy at $T = 50$     (b) *Top-50* accuracy at $T = 0$     (c) *Top-100* accuracy at $T = 0$

Figure 4: Accuracy comparison of sampling w/ DWG and w/o DWG. Here, class 1~4 represent {card about to expire, apple pay or google pay, terminate account and verify source of funds}, respectively.

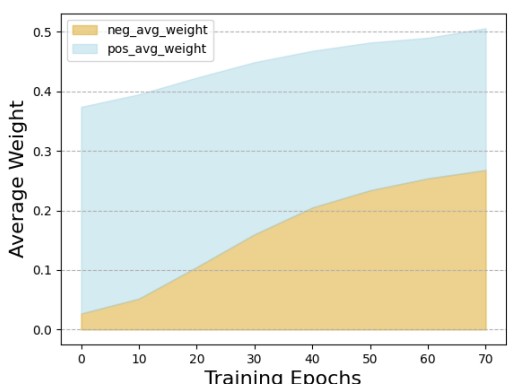

Figure 5: Average weight changing of sampled keys.

and temperature $\tau$ are set to 0.3, 1.1, 0.3 and 0.2, respectively. We adopt the data augmentation of random token replacement as (Zhang et al., 2022). We set the learning rate to 1e-5 and train until convergence without early-stopping. During inference with GSF, we set the number of stacking layers $t$ to 2 and neighborhood size to one-third of the average size of the testing set for each class. All the experiments are conducted on a single RTX-3090 and averaged over 3 runs.

### 4.5 Main Results

The main results are shown in Table 2. Our method outperforms various comparison methods consistently and achieves clustering accuracy improvements of 2.56%, 0.90% and 3.25% on three benchmark datasets compared with previous state-of-the-art CLNN, respectively. It demonstrates the effectiveness of our method to discover new intents with limited known intent data.

## 5 Discussion

### 5.1 Ablation Study

To investigate the contributions of different components in our method, we remove GSF, self-

training and contrastive learning based on DWG in sequence to conduct experiments on BANKING again. As shown in Table 3, removing them impairs model performance consistently, indicating GSF really alleviates the negative effect of high-frequency noise, and both local and global supervision provided by Eq.10 benefit new intent discovery, especially DWG contrastive learning.

### 5.2 Analysis of DWG

To validate the effectiveness of DWG contrastive learning, we compare the model performance under diffusion and non-diffusion conditions. Moreover, we also explore the sensitivity of our method to hyperparameter changes, including the first-order neighborhood size $k$ and the number of diffusion rounds $r$. As shown in Fig.3, DWG generally helps the model outperform the original non-diffusion method adopted by (Zhang et al., 2022) and dramatically reduces the search scope of $k$, indicating our method is both effective and robust.

To further illustrate the positive effect brought by structure relationships, we separately analyze the sampling strategy and weighting strategy based on DWG.

**Sampling Strategy**. Taking the BANKING dataset as an example, we choose 4 representative classes from it according to the sampling difficulty. Fig.4a and Fig.4b show the *Top-50* positive keys sampling accuracy at epoch 50 and 0, respectively, indicating the connectivity required by structure relationships can effectively improve sampling accuracy, especially 1) on categories with high sampling difficulty; 2) at the beginning of the training that samples haven't form compact clusters. Fig.4c shows the *Top-100* sampling accuracy at epoch 0, which indicates our method is more robust to retrieve positive keys selectively when relaxing the threshold.

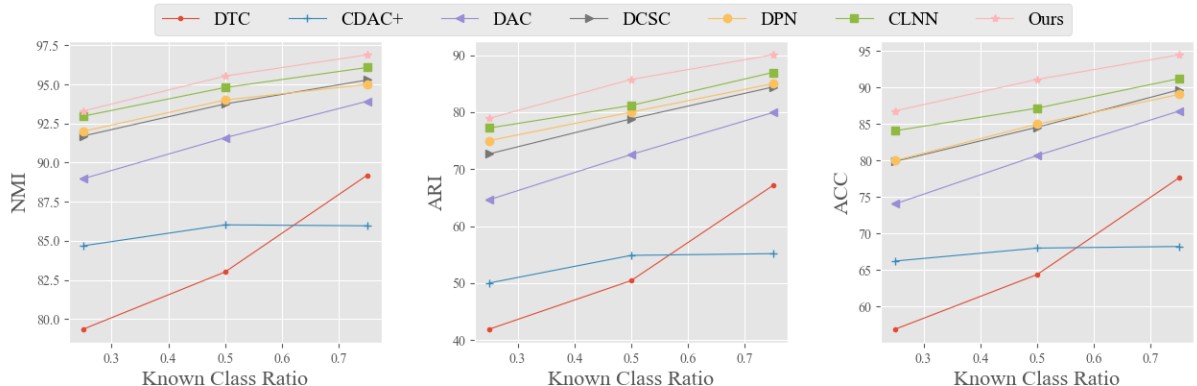

Figure 6: Influence of known class ratio on the CLINC dataset.

|  |  | ARI | ACC | SC |
|---|---|---|---|---|
| w/o GSF | | 88.48 | 92.84 | 0.64 |
| t=1 | k=5 | 88.57 | 92.89 | 0.70 |
| | k=10 | 90.34 | 94.49 | 0.76 |
| | k=15 | 90.05 | 94.36 | 0.74 |
| t=2 | k=5 | 89.95 | 94.31 | 0.72 |
| | k=10 | 89.47 | 94.04 | 0.81 |
| | k=15 | 87.71 | 92.71 | 0.79 |

Table 4: Ablation study on hyperparameters of GSF.

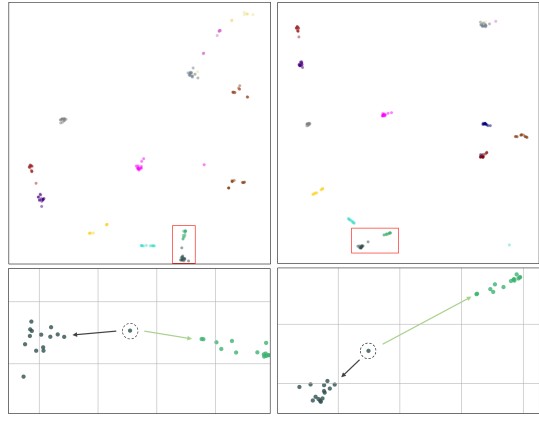

Figure 7: Visualization of embeddings on CLINC. **Left**: w/o GSF. **Right**: w/ GSF.

**Weighting Strategy**. The average weight of sampled positive and negative keys without semantic similarity threshold is presented in Fig.5 per 10 epochs. It clearly shows that positive keys dominate model training consistently, while semantic-unrelated negative keys are suppressed, and the semantic-related negative keys provide rich semantics for training through soft weighting.

### 5.3 Analysis of GSF

To verify the effectiveness of GSF under different stacking layers and neighborhood sizes, we freeze the trained model and perform KMeans clustering with representations smoothed to varying degrees. Table 4 shows the results of ARI, ACC and Silhouette Coefficient (SC) on CLINC. The performance on different evaluation metrics is mostly superior to direct clustering and robust to hyperparameter changes. In particular, the SC value shows a significant improvement, indicating a reduction in clustering uncertainty.

To further illustrate how GSF improves inference, we randomly sample 15 classes from CLINC and t-SNE visualize them. Fig.7 clearly shows the more compact cluster distributions after smoothing, and the partially zoomed-in illustrations show that

GSF corrects some semantically ambiguous samples on the boundary by bringing them closer to the side with stronger connectivity.

### 5.4 Influence of Known Class Ratio

To investigate the influence of the known class ratio on model performance, we vary it in the range of 0.25, 0.50 and 0.75. As shown in Fig.6, our method achieves comparable or best performance under different settings on all evaluation metrics, which fully demonstrates the effectiveness and robustness of our method.

### 6 Conclusion

In this paper, we propose a novel Diffusion Weighted Graph Framework (DWGF) for new intent discovery, which models structure relationships inherent in data through nearest neighbor-guided diffusion. Combined with structure relationships, we improve both the sampling and weighting strategy in contrastive learning and adopt supervision from local and global views. We further

propose Graph Smoothing Filter (GSF) to explore the potential of structure relationships in inference, which effectively filters noise embodied in semantically ambiguous samples on the cluster boundary. Extensive experiments on all three clustering metrics across multiple benchmark datasets fully validate the effectiveness and robustness of our method.

## Limitations

Even though the proposed Diffusion Weighted Graph framework achieves superior performance on the NID task, it still faces the following limitations. Firstly, the construction of DWG and GSF needs extra hyperparameters, and their changes will slightly impact the model's performance. Secondly, it is time-consuming to do nearest neighbor retrieval on the entire dataset.

## Acknowledgments

This work was supported by National Key Research and Development Program of China (2022ZD0117102), National Natural Science Foundation of China (62293551, 62177038, 62277042, 62137002, 61721002, 61937001, 62377038). Innovation Research Team of Ministry of Education (IRT_17R86), Project of China Knowledge Centre for Engineering Science and Technology, "LENOVO-XJTU" Intelligent Industry Joint Laboratory Project.

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
