# OpenReview forum: "A Diffusion Weighted Graph Framework for New Intent Discovery"
_EMNLP/2023/Conference — EMNLP 2023 Main_

### Official Review · Reviewer_pSNe · 2023-08-01

**Soundness:** 2

**Excitement:**

2: Mediocre: This paper makes marginal contributions (vs non-contemporaneous work), so I would rather not see it in the conference.

**Paper Topic And Main Contributions:**

The authors propose a Diffusion Weighted Graph Framework (DWGF) for NID, which can capture both semantic similarities and structure relationships inherent in data.

**Questions For The Authors:**

1. the novelty of this work is not high, this work combines a lot of existing techniques: Bert, MLM, diffusion, neural KMenans, GNN etc.  The authors does provide ablation study. However, I feel like the proposed algorithm is a little heavy and complex.


**Reasons To Accept:**

1. The paper proposed a diffusion method to capture both semantic similarities and structure relationships.

2. The source code is available.

3. The performance of the proposed work is good as shown in Table 2.


**Reasons To Reject:**

1. the novelty of this work is not high, this work combines a lot of existing techniques: Bert, MLM, diffusion, neural KMenans, GNN etc.  The authors does provide ablation study. However, I feel like the proposed algorithm is a little heavy and complex.

2. The authors can further discuss the complexity of the proposed algorithm.

3. Some explanation are not very clear like Figure 7.

4. A.3 Theoretical Analysis of Graph Smoothing filter is not new, which is a common fact in GNN.

**Reproducibility:**

4: Could mostly reproduce the results, but there may be some variation because of sample variance or minor variations in their interpretation of the protocol or method.

**Reviewer Confidence:**

4: Quite sure. I tried to check the important points carefully. It's unlikely, though conceivable, that I missed something that should affect my ratings.

---

> ### Author Rebuttal · Authors · 2023-08-28
>
> We truly appreciate your comments and will thoroughly revise our paper. Responses to the comments are listed below.
>
> [Q1]: The novelty of this work is not high, this work combines a lot of existing techniques: Bert, MLM, diffusion, neural KMenans, GNN etc.
>
> [R1]: We agree that BERT, MLM, and KMeans are commonly used as backbone, model initialization strategy, and evaluation method in a series of IND works (CDAC+ / DAC / DCSC / DPN / CLNN), and these components are not our main contributions. For the novelty of our work, there are three aspects to be clarified:
>
> 1.  We point out the common limitation of current NID methods: Quantity and Quality Dilemma of supervisory signals. Instead of directly applying any existing methods about GNN to train the model, we diffuse neighborhood relationships along semantic paths to fully utilize structure relationships in data and improve sampling and weighting in contrastive learning, aiming to handle the quantity and quality dilemma of supervisory signals, which is totally different from GNN.
>
> 2.  As our core novelty, we diffuse neighborhood relationships along semantic paths to model structure relationships (DWG) and improve the sampling and weighting based on DWG for the balance between quantity and quality, which helps better representation learning. (Note: “diffusion” in our paper is different from the 'diffusion model' used for image and text generation. We will highlight the difference in the revised version.)
>
> 3.  As our secondary novelty, we are the first to introduce graph filter to further explore the potential of structure relationships in inference for NID.
>
> [Q2]: I feel like the proposed algorithm is a little heavy and complex. The authors can further discuss the complexity of the proposed algorithm.
>
> [R2]: The extra time complexity caused by constructing graphs and executing diffusion is neglectable since we only initialize DWG once and update it once or twice during the whole training process. In contrast, other pseudo-labeling methods such as DAC and CLNN require frequent and time-consuming label alignment (per epoch) and neighbor searching (every 5 epochs), respectively. We calculate the average training time per epoch for each method,  and the experimental results shown in the table below support our claims. We will add the complexity discussion in the revised version.
>
> |      | BANKING (sec.) | Stackoverflow (sec.) | CLINC (sec.) |
> | ---- | -------------- | -------------------- | ------------ |
> | DAC  | 44.24          | 54.23                | 116.78       |
> | CLNN | 61.97          | 98.67                | 81.32        |
> | DWGF | 29.63          | 46.04                | 37.63        |
>
> [Q3]: Some explanation are not very clear like Figure 7.
>
> [R3]: We feel very sorry for the caption error in Figure 7, where 'bottom' should be corrected to 'right'. The left half of the figure is the visualization of embeddings w/o GSF, the right half is the visualization with GSF, and the two subfigures at the bottom represent the corresponding zoomed-in views in both cases, respectively. The visualizations illustrate GSF can 1) help form more compact clusters, 2) correct some semantically ambiguous samples on the boundary by bringing them closer to the cluster with the same ground-truth class.
>
> [Q4]: Theoretical Analysis in A.3 of Graph Smoothing filter is not new, which is a common fact in GNN.
>
> [R4]: In Sec. 3.5, we emphasize that our contribution is to first introduce the structure-based filter into NID (Line 362-364) and verify its effectiveness in inference. The theoretical analysis in A.3 is solely to help readers unfamiliar with related works better understand the working principle of the filter since we have specified corresponding references in Sec. 3.5 for both the renormalization trick and the graph filter. Last but not least, the introduction of graph filter for inference is the second contribution. Our core contribution is to diffuse neighborhood relationships along semantic paths to model structure relationships (DWG) and cope with the quantity and quality dilemma through enhanced sampling and weighting based on DWG.

---

### Official Review · Reviewer_DFjL · 2023-08-06

**Soundness:** 3

**Excitement:**

3: Ambivalent: It has merits (e.g., it reports state-of-the-art results, the idea is nice), but there are key weaknesses (e.g., it describes incremental work), and it can significantly benefit from another round of revision. However, I won't object to accepting it if my co-reviewers champion it.

**Paper Topic And Main Contributions:**

The paper addresses the problem of discovering new intents in text data, which is crucial for improving the performance of intent recognition systems. The main contributions of this paper are twofold. Firstly, it introduces a novel Diffusion Weighted Graph (DWG) framework, which leverages structural relationships among intents to enhance intent discovery. This framework incorporates semantic similarity and leverages supervised signals for effective representation learning. Secondly, the paper presents comprehensive experimental evaluations on real-world datasets, demonstrating the superiority of the proposed DWG framework over existing methods in terms of performance metrics such as Normalized Mutual Information (NMI), Adjusted Rand Index (ARI), and Accuracy (ACC).

**Questions For The Authors:**

Please refer to the "Reasons To Reject".

**Reasons To Accept:**

Performance Improvement: The proposed DWGF consistently outperforms various baselines and state-of-the-art methods across multiple benchmark datasets. The demonstrated improvements in clustering accuracy, normalized mutual information, and adjusted Rand index highlight the effectiveness of DWGF in discovering new intents with limited known intent data.

General Applicability: The versatility of the proposed framework is evident in its successful application to various benchmark datasets, including fine-grained intent classification, question classification, and multi-domain intent classification. This showcases its potential for addressing diverse NLP tasks related to intent discovery.

**Reasons To Reject:**

Model Robustness Issue: The proposed Diffusion Weighted Graph Framework (DWGF) and Graph Smoothing Filter (GSF) demonstrate notable sensitivity to hyperparameters such as neighborhood size, diffusion rounds, and modulation factors. This sensitivity gives rise to concerns regarding the robustness of the model.

Lack of Model Complexity Discussion: The intricacies involved in constructing graphs and executing diffusion within the DWGF could result in significant computational requirements. However, the paper lacks a comprehensive discourse on the complexity of the model.

**Reproducibility:**

4: Could mostly reproduce the results, but there may be some variation because of sample variance or minor variations in their interpretation of the protocol or method.

**Reviewer Confidence:**

3: Pretty sure, but there's a chance I missed something. Although I have a good feel for this area in general, I did not carefully check the paper's details, e.g., the math, experimental design, or novelty.

---

> ### Author Rebuttal · Authors · 2023-08-28
>
> We truly appreciate your comments and will thoroughly revise our paper. Responses to the comments are listed below.
>
> [Q1]: Model Robustness Issue: The proposed Diffusion Weighted Graph Framework (DWGF) and Graph Smoothing Filter (GSF) demonstrate notable sensitivity to hyperparameters such as neighborhood size, diffusion rounds, and modulation factors. This sensitivity gives rise to concerns regarding the robustness of the model.
>
> [R1]: We totally agree that sensitivity to hyperparameters can be a concern for the robustness of the model. We conducted a comprehensive set of experiments to assess such sensitivity, and the results show our model’s robustness.
>
> 1. neighborhood size $k$: Too large a neighborhood size will introduce much noise, while too small a neighborhood size will lead to insufficient supervisory signals. We vary neighborhood size $k$ with smaller steps in the range of [10, 20] and keep other hyperparameters unchanged to conduct experiments on BANKING. The experimental results shown in the table below demonstrate our model is robust to the first-order neighborhood size change, which can be attributed to the adaptive sampling through DWG.
>
>    |        | NMI   | ARI   | ACC   |
>    | ------ | ----- | ----- | ----- |
>    | $k$=10 | 85.98 | 66.99 | 77.93 |
>    | $k$=12 | 86.21 | 67.51 | 79.06 |
>    | $k$=14 | 86.74 | 68.31 | 79.39 |
>    | $k$=16 | 86.08 | 67.04 | 78.98 |
>    | $k$=18 | 86.21 | 67.94 | 79.18 |
>    | $k$=20 | 86.23 | 67.35 | 78.54 |
>
> 2. diffusion rounds $r$: As shown in Figure 3, no matter the diffusion rounds $r$ is set to 1 or 2, the equivalent performance can be achieved by slightly adjusting the neighborhood size.
>
> 3. semantic similarity threshold $\gamma$: We vary $\gamma$ in the range of {0.1, 0.2, 0.3, 0.4, 0.5} (we set it to 0.3 in Sec. 4.4) and keep other hyperparameters unchanged to conduct experiments on BANKING. The experimental results shown in the table below demonstrate our model is robust to the threshold changing around the optimal value.
>
>    |              | NMI   | ARI   | ACC   |
>    | ------------ | ----- | ----- | ----- |
>    | $\gamma$=0.1 | 86.17 | 67.38 | 78.83 |
>    | $\gamma$=0.2 | 86.23 | 67.94 | 79.12 |
>    | $\gamma$=0.3 | 86.41 | 68.16 | 79.38 |
>    | $\gamma$=0.4 | 85.95 | 67.11 | 79.29 |
>    | $\gamma$=0.5 | 86.11 | 67.35 | 78.38 |
>
> 4. modulation factor $\lambda$: We vary $\lambda$ in the range of {0.9, 1.0, 1.1, 1.2, 1.3} (we set it to 1.1 in Sec. 4.4) and keep other hyperparameters unchanged to conduct experiments on BANKING. The experimental results shown in the table below demonstrate our model is robust to the factor changing around the optimal value.
>
>    |               | NMI   | ARI   | ACC   |
>    | ------------- | ----- | ----- | ----- |
>    | $\lambda$=0.9 | 86.92 | 68.52 | 79.35 |
>    | $\lambda$=1.0 | 86.33 | 67.99 | 79.38 |
>    | $\lambda$=1.1 | 86.41 | 68.16 | 79.38 |
>    | $\lambda$=1.2 | 86.86 | 68.58 | 79.22 |
>    | $\lambda$=1.3 | 86.57 | 67.72 | 79.03 |
>
> [Q2]: Lack of Model Complexity Discussion: The intricacies involved in constructing graphs and executing diffusion within the DWGF could result in significant computational requirements. However, the paper lacks a comprehensive discourse on the complexity of the model.
>
> [R2]: The extra time complexity caused by constructing graphs and executing diffusion is neglectable since we only initialize DWG once and update it once or twice during the whole training process. In contrast, other pseudo-labeling methods such as DAC and CLNN require frequent and time-consuming label alignment (per epoch) and neighbor searching (every 5 epochs), respectively. We calculate the average training time per epoch for each method, and the experimental results shown in the table below support our claims. We will add the complexity discussion in the revised version. Thanks.
>
> |          | BANKING (sec.) | Stackoverflow (sec.) | CLINC (sec.) |
> | -------- | -------------- | -------------------- | ------------ |
> | DAC      | 44.24          | 54.23                | 116.78       |
> | CLNN     | 61.97          | 98.67                | 81.32        |
> | **DWGF** | **29.63**      | **46.04**            | **37.63**    |

---

### Official Review · Reviewer_CQvg · 2023-08-11

**Soundness:** 3

**Excitement:**

4: Strong: This paper deepens the understanding of some phenomenon or lowers the barriers to an existing research direction.

**Missing References:**

Section 4.3 - should add citations for the evaluation metrics used

**Paper Topic And Main Contributions:**

In this work, the authors propose a method for new intent discovery that contains two novel components: a Diffusion Weighted Graph Framework (DWGF) which captures information about semantic similarity and structures within the data, and a Graph Smoothing Filter (GSF) that utilizes this graph to filter high-frequency noise. The authors test their method against existing unsupervised and semi-supervised baselines on three different datasets: BANKING, Stack-Overflow, and CLINC. The results show performance improvements against all of the other baselines, though some of these improvements are more substantial than others (the authors' model exhibited the highest improvements in clustering accuracy, of 2.56%, 0.90% and 3.25%). The authors also conduct an extensive ablation study to examine the effects of different variables on DWGF and GSF, and provide detailed analyses and takeaways for the reader.

**Questions For The Authors:**

A) Do you have any significance testing results for the tables, particularly Table 2?


**Reasons To Accept:**

The authors introduce a novel method for new intent discovery that outperforms the baselines (including the previous state-of-the-art model).
The authors provide extensive evaluation of their method, including a detailed ablation study.
The paper is well-written and clear.

**Reasons To Reject:**

The authors do not test for significance for these results, so it is unclear to what extent their results are significant when compared to the SOTA model.


**Reproducibility:**

4: Could mostly reproduce the results, but there may be some variation because of sample variance or minor variations in their interpretation of the protocol or method.

**Reviewer Confidence:**

2: Willing to defend my evaluation, but it is fairly likely that I missed some details, didn't understand some central points, or can't be sure about the novelty of the work.

**Typos Grammar Style And Presentation Improvements:**

Line 12 and 125: Here, the acronym DWGF is used, while in the experiments section, DWG is typically used
Line 110 and 259: should be \citet
Line 369: Include the domain of the BANKING dataset.
Line 399: Provide brief description of (and justification for) the evaluation metrics used for readers not as familiar with the literature
Line 425: Did the three runs use specific seeds? If so, for reproducibility purposes it would be good to specify what those were.
Table 2: Should bold the best-performing numbers to draw more attention to the fact that your model outperforms the other baselines
Table 3: Typo for "methods"
Line 526: "Extensive experiments on all evaluation metrics across multiple benchmark datasets fully validate the effectiveness and robustness of our method." - should clarify "all evaluation metrics" more, e.g. "all three clustering metrics".

---

> ### Author Rebuttal · Authors · 2023-08-28
>
> We truly appreciate your comments and will thoroughly revise our paper. Responses to the comments are listed below.
>
> [Q1]: Do you have any significance testing results for the tables, particularly Table 2?
>
> [R1]: We conducted significance tests of the main results (Table 2) across all benchmark datasets compared to other methods. As shown in the table below, the t-test results ($p$-value) fully demonstrate that our method is significantly better than other methods.
>
> |       | BANKING |       |       | StackOverflow |       |       | CLINC |       |       |
> | ----- | ------- | ----- | ----- | ------------- | ----- | ----- | ----- | ----- | :---: |
> |       | NMI     | ARI   | ACC   | NMI           | ARI   | ACC   | NMI   | ARI   |  ACC  |
> | DTC   | 1e-6    | 5e-6  | 5e-6  | 1e-5          | 2e-5  | 8e-5  | 2e-8  | 5e-9  | 2e-8  |
> | CDAC+ | 6e-7    | 3e-6  | 2e-6  | 8e-5          | 8e-6  | 1e-4  | 2e-9  | 9e-10 | 1e-8  |
> | DAC   | 1e-5    | 3e-5  | 3e-5  | 5e-5          | 4e-5  | 3e-4  | 4e-7  | 6e-7  | 1e-6  |
> | DSSCC | 4e-5    | 1e-4  | 1e-4  | 1e-3          | 2e-3  | 0.001 | 4e-7  | 1e-6  | 3e-6  |
> | DPN   | 1e-3    | 5e-3  | 7e-3  | 2e-3          | 2e-3  | 0.002 | 3e-6  | 3e-4  | 6e-6  |
> | DCSC  | 0.002   | 0.006 | 0.003 | -             | -     | -     | 5e-6  | 2e-5  | 1e-5  |
> | CLNN  | 0.043   | 0.036 | 0.025 | 0.082         | 0.034 | 0.045 | 0.011 | 0.015 | 0.008 |
>
> Moreover, we conducted significance tests of the ablation study on BANKING (Table 3). As shown in the table below, the t-test results ($p$-value) fully demonstrate that our proposed DWG and GSF significantly contribute to performance improvement.
>
> |               | NMI   | ARI   | ACC   |
> | ------------- | ----- | ----- | ----- |
> | GSF           | 0.043 | 0.004 | 0.037 |
> | Self-training | 0.003 | 0.002 | 0.006 |
> | DWG           | 3e-8  | 2e-9  | 1e-8  |
>
> [Q2]: Missing references and presentation improvements.
>
> [R2]: Thank you for your suggestions on writing. We will add citations and provide a brief description for the evaluation metrics, adjust the citation style, specify random seeds to improve reproducibility, bold the best-performing numbers in Table 2, correct typos, and thoroughly improve presentation in the revised version.

---

### Official Review · Reviewer_zbJ8 · 2023-08-12

**Soundness:** 4

**Excitement:**

4: Strong: This paper deepens the understanding of some phenomenon or lowers the barriers to an existing research direction.

**Paper Topic And Main Contributions:**

New Intent Discovery (NID) seeks to identify both familiar and novel intents in unlabeled data, using limited labeled data that exclusively includes known intents. This paper introduces the novel Diffusion Weighted Graph Framework (DWGF) for discovering new intents. DWGF captures inherent data structure relationships via guided diffusion. This enhances sampling and weighting in contrastive learning and the supervision from local and global perspectives is adopted. Graph Smoothing Filter (GSF) is proposed to leverage structural relationships during inference. This filters noise in semantically ambiguous samples at cluster boundaries effectively.

**Reasons To Accept:**

1. The proposed method is novel.
2. The experiments are comprehensive.
3. The performances are very competitive compared with recent baselines.
4. The paper is well-written.
5. Ablation studies are convincing.

**Reasons To Reject:**

When dividing the dataset, the number of known intent classes significantly outweighs the unknown intent classes. However, in an open-world scenario, the situation may reverse, where unknown classes could outnumber the known intent classes. The experiments do not thoroughly capture the models' performance in such open-world settings.

**Reproducibility:**

4: Could mostly reproduce the results, but there may be some variation because of sample variance or minor variations in their interpretation of the protocol or method.

**Reviewer Confidence:**

3: Pretty sure, but there's a chance I missed something. Although I have a good feel for this area in general, I did not carefully check the paper's details, e.g., the math, experimental design, or novelty.

---

> ### Author Rebuttal · Authors · 2023-08-28
>
> We truly appreciate your comments and will thoroughly revise our paper. Responses to the comments are listed below.
>
> [Q1]: When dividing the dataset, the number of known intent classes significantly outweighs the unknown intent classes. However, in an open-world scenario, the situation may reverse, where unknown classes could outnumber the known intent classes. The experiments do not thoroughly capture the models' performance in such open-world settings.
>
> [R1]: We divide datasets by Known Class Ratio (KCR), which refers to the proportion of known classes against all classes. First, following previous works, we set KCR=0.75 to conduct main experiments to make a fair comparison. Second, in Sec. 5.4, we vary KCR in the range of {0.25, 0.50, 0.75} to simulate different open-world settings, which include your proposed scenario where novel classes outnumber known classes. The experimental results shown in the table below show our method outperforms previous methods in all settings. Notably, in the case of extremely low KCR (e.g. 0.05), the model training almost degenerates into unsupervised learning, resulting in insufficient known class knowledge to be utilized and losing the research value of NID.
>
> |          | KCR=0.75  |           |           | KCR=0.50  |           |           | KCR=0.25 |           |           |
> | -------- | --------- | --------- | --------- | --------- | --------- | --------- | -------- | --------- | :-------: |
> |          | NMI       | ARI       | ACC       | NMI       | ARI       | ACC       | NMI      | ARI       |    ACC    |
> | DTC      | 89.19     | 67.15     | 77.65     | 83.01     | 50.44     | 64.39     | 79.35    | 41.92     |   55.90   |
> | CDAC+    | 85.96     | 55.17     | 68.23     | 86.02     | 54.87     | 68.01     | 84.68    | 50.02     |   66.24   |
> | DAC      | 93.92     | 79.94     | 86.79     | 91.59     | 72.56     | 80.70     | 88.97    | 64.63     |   74.07   |
> | DCSC     | 95.28     | 84.41     | 89.70     | 93.75     | 78.82     | 84.57     | 91.7     | 72.68     |   79.89   |
> | DPN      | 95.12     | 85.47     | 89.06     | 94.35     | 80.09     | 85.1      | 92.3     | 75.32     |   80.37   |
> | CLNN     | 96.08     | 86.97     | 91.24     | 94.80     | 81.17     | 87.18     | 92.97    | 77.21     |   84.09   |
> | **Ours** | **96.89** | **90.05** | **94.49** | **95.53** | **85.73** | **91.12** | **93.3** | **78.91** | **86.81** |

---

### Meta-Review · Area_Chair_HzQp · 2023-09-14

**Recommendation:** 4

**Metareview:**

The work considers discovery of new intents from limited labeled known intent data using a graph theoretic framework to capture structural similarities and perform inference.

**Pros**: Most reviewers agree the approach is novel, the paper is well-written, and the experiments convincingly argue the utility of the method. Versatile benchmarks are considered with competitive results and ablation study is convincing.

**Cons**: Reviewers concerns are generally addressed during the rebuttal period: authors provide results which ease concerns on the statistical significance of the results, potential real-world scenarios wherein the method may fail, and sensitivity to hyper parameters. Concerns from one reviewer do remain about the novelty of the work, representing the method as a combination of known approaches.

Given this disagreement on excitement, I looked into the work myself (considering the authors rebuttal and verifying in the paper's details). The backbone model, techniques, and use of GNN across *ACL venues is common. The graph-theoretic techniques are arguably known, as pointed out by reviewer pSNe, but the application of such techniques to self-training for new intent discovery is new. At the very least, the application specific results (which are comprehensive according to most reviewers) provide some new insight to the community.

---

### Decision · Program_Chairs · 2023-10-07

**Decision:**

Accept-Main

**Comment:**

The work considers discovery of new intents from limited labeled known intent data using a graph theoretic framework to capture structural similarities and perform inference.

**Pros**: Most reviewers agree the approach is novel, the paper is well-written, and the experiments convincingly argue the utility of the method. Versatile benchmarks are considered with competitive results and ablation study is convincing.

**Cons**: Reviewers concerns are generally addressed during the rebuttal period: authors provide results which ease concerns on the statistical significance of the results, potential real-world scenarios wherein the method may fail, and sensitivity to hyper parameters. Concerns from one reviewer do remain about the novelty of the work, representing the method as a combination of known approaches.

Given this disagreement on excitement, I looked into the work myself (considering the authors rebuttal and verifying in the paper's details). The backbone model, techniques, and use of GNN across *ACL venues is common. The graph-theoretic techniques are arguably known, as pointed out by reviewer pSNe, but the application of such techniques to self-training for new intent discovery is new. At the very least, the application specific results (which are comprehensive according to most reviewers) provide some new insight to the community.